# Fecal Microbiota Transplant in a Pre-Clinical Model of Type 2 Diabetes Mellitus, Obesity and Diabetic Kidney Disease

**DOI:** 10.3390/ijms23073842

**Published:** 2022-03-31

**Authors:** Rosana M. C. Bastos, Antônio Simplício-Filho, Christian Sávio-Silva, Luiz Felipe V. Oliveira, Giuliano N. F. Cruz, Eliza H. Sousa, Irene L. Noronha, Cristóvão L. P. Mangueira, Heloísa Quaglierini-Ribeiro, Gleice R. Josefi-Rocha, Érika B. Rangel

**Affiliations:** 1Hospital Israelita Albert Einstein, São Paulo 05652-900, SP, Brazil; rmcbastos22@gmail.com (R.M.C.B.); antonio.simplicio@einstein.br (A.S.-F.); christian.silvio@einstein.br (C.S.-S.); eliza.higuti@einstein.br (E.H.S.); cristovao.mangueira@einstein.br (C.L.P.M.); heloisa.quaglierini@gmail.com (H.Q.-R.); gjosefi@gmail.com (G.R.J.-R.); 2BiomeHub, Santa Catarina 88040-900, SC, Brazil; felipe@biome-hub.com (L.F.V.O.); giuliano.netto@biome-hub.com (G.N.F.C.); 3Division of Nephrology, School of Medicine, University of São Paulo, São Paulo 01246-903, SP, Brazil; irenenor@usp.br; 4Nephrology Division, Federal University of São Paulo, São Paulo 04023-900, SP, Brazil

**Keywords:** diabetes mellitus, diabetic kidney disease, gut microbiota, fecal transplant, obesity

## Abstract

Diabetes mellitus (DM) burden encompasses diabetic kidney disease (DKD), the leading cause of end-stage renal disease worldwide. Despite compelling evidence indicating that pharmacological intervention curtails DKD progression, the search for non-pharmacological strategies can identify novel targets for drug development against metabolic diseases. One of those emergent strategies comprises the modulation of the intestinal microbiota through fecal transplant from healthy donors. This study sought to investigate the benefits of fecal microbiota transplant (FMT) on functional and morphological parameters in a preclinical model of type 2 DM, obesity, and DKD using BTBR^ob/ob^ mice. These animals develop hyperglycemia and albuminuria in a time-dependent manner, mimicking DKD in humans. Our main findings unveiled that FMT prevented body weight gain, reduced albuminuria and tumor necrosis factor-α (TNF-α) levels within the ileum and ascending colon, and potentially ameliorated insulin resistance in BTBR^ob/ob^ mice. Intestinal structural integrity was maintained. Notably, FMT was associated with the abundance of the succinate-consuming *Odoribacteraceae* bacteria family throughout the intestine. Collectively, our data pointed out the safety and efficacy of FMT in a preclinical model of type 2 DM, obesity, and DKD. These findings provide a basis for translational research on intestinal microbiota modulation and testing its therapeutic potential combined with current treatment for DM.

## 1. Introduction

Diabetic kidney disease (DKD) is one of the major complications of diabetes mellitus (DM) [1,2] and develops in approximately 40% of diabetic individuals [1,3]. DKD is considered the main cause of end-stage kidney disease (ESKD) [3], resulting in increased morbidity and mortality rates, mainly due to cardiovascular complications [2].

Several risk factors contribute to the development and progression of DKD, including non-modifiable and modifiable variables, such as age, sex, ethnicity, family history of DM, duration of DM, glycemic control, hypertension, and obesity [1,3]. The pathogenesis of DKD is complex and is triggered by the exposure of renal cells to excessive glucose inflow. That hyper glucose burden leads to metabolic stress, hemodynamic alterations, overactive renin-angiotensin-aldosterone system (RAAS), and upregulation of various signaling pathways resulting in oxidative stress, cellular dysfunction, inflammation, apoptosis and fibrosis [1]. Albuminuria is defined by a urinary albumin-to-creatinine ratio (UACR) ≥ 30 mg/g [1] and is the hallmark of DKD diagnosis and progression [2,3]. It is associated with an increased risk of ESKD and cardiovascular disease [2,3]. Currently, therapeutic strategies to prevent DKD comprise adequate glycemic control, dietary and lifestyle interventions, dyslipidemia treatment, blood pressure control, blockade of RAAS, and antidiabetic drugs [1,2]. Despite broad pharmacological and non-pharmacological approaches, DKD has evolved throughout the years, which prompted the development of novel strategies to curtail its progression.

Among these novel therapeutic approaches, the modulation of the intestinal microbiota has emerged as a promising strategy. The gut microbiota is a complex microbial community that plays an important role in many aspects of human health [4]. Several factors can influence the composition of that microbiota, mode of birth, infant type of feeding, host genetics, age, diet, lifestyle, medication, burden of comorbidities, and environmental exposures [4]. The imbalance in the composition and function of the gut microbiota has been associated with metabolic disorders, including obesity, insulin resistance, and type 2 DM (T2DM) [5,6,7]. The gut microbiota participates in energy homeostasis, glucose metabolism, insulin resistance, and weight gain [5,8,9]. Moreover, gut microbiota-derived lipopolysaccharide (LPS) produced by Gram-negative bacteria is involved in compromising the integrity of the intestinal barrier, which leads to subclinical inflammation and DM burden [9,10]. Diabetic individuals are dysbiotic and present a less diverse [7], dysfunctional, depleted in butyrate-producing bacteria microbiota, as well as an increase in opportunistic pathogens [6], Likewise, DM is associated with a lower abundance of the *Firmicutes* phylum, whereas higher proportions of the *Bacteroidetes* and *Proteobacteria* phyla and *Betaproteobacteria* class were found [11]. Thus, understanding human microbiome homeostasis is of paramount importance for therapeutic purposes, including dietary interventions [12], use of prebiotics [13], probiotics [14], symbiotics [15], and fecal microbiota transplant (FMT) [16,17,18]. FMT consists of the administration of a solution of fecal matter from healthy donors in order to modify the gut microbial composition of unhealthy recipients [18]. Here, we hypothesized that FMT may have beneficial effects on functional and morphological parameters in the preclinical model of DKD using the BTBR^ob/ob^ mice, the most robust animal model that empresses the human features of DKD [19].

## 2. Results

### 2.1. 16S rRNA Sequencing Analysis

In BTBR (Black and Tan, Obese Tufted) wild-type mice (WT), BTBR^ob/ob^ mice not submitted to FMT or FMT (-), and BTBR^ob/ob^ mice submitted to FMT or FMT (+), sequencing of the 16S rRNA disclosed seven bacterial phyla, including *Bacteroidetes*, *Firmicutes*, *Proteobacteria*, *Actinobacteria*, *Verrucomicrobia*, *Deferribacteres*, and *Chloroflex* (Figure 1A). In all groups, most sequences revealed greater proportions of the *Bacteroidetes* phylum when compared to the *Firmicutes* phylum. See Appendix A for detailed information.

Alpha-diversity analysis using Shannon’s index indicated that the species richness was similar among groups (Figure 1B). Beta diversity analysis, using principal coordinate analysis (PCoA) based on Bray–Curtis dissimilarity metrics, was not associated with the treatment (Figure 1C).

When we evaluated the rate of butyrate-producing bacteria (*Ruminococcaceae* and *Lachnospiraceae*) relative to non-producing bacteria, an increase in the proportion in the *Lachnospiraceae* family was observed in 14-week-old BTBR WT mice over 14-week-old BTBR^ob/ob^ FMT (-) and FMT (+) mice with similar ages (*p* = 0.026) (Figure 1D). Next, we investigated the effect of genotype and treatment on the microbiota quantified by a log2-fold change and found a higher relative abundance in the *Gammaproteobacteria* and *Verrucomicrobiae* classes and a lower abundance in the *Dehalococcoidia* and *Odoribacteraceae* families in 14-week-old BTBR^ob/ob^ FMT (-) mice. Importantly, FMT increased the abundance of the *Odoribacteraceae* family in 14-week-old BTBR^ob/ob^ mice (Figure 1E).

### 2.2. Functional and Metabolic Parameters in BTBR WT, BTBR^ob/ob^ FMT (-), and BTBR^ob/ob^ FMT (+) Mice

As expected, BTBR^ob/ob^ mice exhibited higher body weight gain when compared to BTBR WT mice (*p* > 0.05), and this difference was maintained over time. However, BTBR^ob/ob^ FMT (+) mice presented lower body weight when compared to untreated BTBR^ob/ob^ mice at all time points (*p* < 0.05) (Figure 2A). All BTBR^ob/ob^ mice, regardless of age and treatment, had significantly increased blood glucose when compared to BTBR WT mice (*p* < 0.05) (Figure 2B).

Glycosuria was higher in 14-week-old BTBR^ob/ob^ FMT (-) and FMT (+) mice when compared to age-matched BTBR WT mice and showed no difference due to treatment (*p* > 0.05) (Figure 2C). Body weight was positively correlated with the *Verrucomicrobia* phylum in 14-week-old BTBR^ob/ob^ FMT (-) mice (r = 0.9, * *p* = 0.02) (Figure 2D) and the *Flavonifractor* genus (r *=* 0.88, * *p =* 0.02) (Figure 2E) and *Betaproteobacteria* class (r = 0.93, ** *p* = 0.008) in BTBR^ob/ob^ FMT (+) mice (Figure 2F).

Regarding metabolic parameters, plasma insulin (Figure 2G) and C-peptide (Figure 2H) levels were higher in the 10- and 14-week-old BTBR^ob/ob^ when compared to age-matched BTBR WT mice (*p* < 0.05), and FMT abrogated that increase. Plasma glucagon was not significantly different between 14-week-old BTBR^ob/ob^ FMT (-) and FMT (+) mice (*p* > 0.05) (Figure 2I). To access insulin resistance and secretion using fasting glucose and insulin concentration, we evaluated HOMA-IR and HOMA-β indexes. Thus, HOMA-IR was higher in 10- and 14-week-old BTBR^ob/ob^ mice when compared to age-matched BTBR WT mice (*p* < 0.05), and FMT prevented the increase in HOMA-IR in BTBR^ob/ob^ mice (Figure 2J). Islet cell function, assessed by HOMA-β, was higher in 10-week-old and BTBR^ob/ob^ mice and was not affected by the treatment (Figure 2K).

### 2.3. Pancreas Histological and Metabolic Parameters and Plasma Enteroendocrine Hormones Evaluation

Morphological evaluation of the pancreatic islet area demonstrated hypertrophy in all BTBR^ob/ob^ mice, regardless of age and treatment, when compared to BTBR WT mice (*p* > 0.05) (Figure 3A).

To gain insights into the secretion of enteroendocrine hormones in DKD and obesity settings, we evaluated glucagon- like peptide -1 (GLP-1 )levels. That hormone exerts a positive impact on glucose and energy homeostasis by stimulating glucose-dependent insulin secretion, as well as inhibiting glucagon release and decreasing food intake [20]. GLP-1 was not significantly different among treated and untreated BTBR^ob/ob^ mice in comparison to BTBR WT mice (*p* > 0.05) yet was significantly higher in 10-week-old BTBR^ob/ob^ mice (Figure 3B). On the other hand, glucagon-like peptide 2 (GLP-2), which exerts an intestinotrophic effect, improving intestinal epithelial proliferation and reducing intestinal permeability [21], was elevated in 14-week-old BTBR^ob/ob^ FMT (-) and FMT (+) mice when compared to age-matched BTBR WT mice (*p* < 0.05), and not affected by FMT (Figure 3C). Peptide YY (PYY), an intestinal hormone associated with delayed gastric emptying, gut motility, and reduced appetite [21,22], was higher in 14-week-old BTBR^ob/ob^ FMT (-) mice, whereas FMT prevented that increase (Figure 3D). Glucose-dependent insulinotropic polypeptide (GIP), an incretin released by nutrients present in the gastrointestinal tract that potentiates insulin secretion [20], was significantly elevated in 14-week-old BTBR^ob/ob^ FMT (-) and FMT (+) mice when compared to age-matched BTBR WT mice (*p* < 0.05) (Figure 3E).

### 2.4. Renal Functional and Histological Parameters and Systemic Inflammation

FMT prevented the increase in urinary albumin-to-creatinine ratio (UACR) in 14-week-old BTBR^ob/ob^ mice (Figure 4A). Despite the impact of FMT on albuminuria, we only observed a slight decrease in hyperfiltration, as assessed by creatinine clearance (Figure 4B). In 14-week-old BTBR^ob/ob^ FMT (+) mice, UACR was associated negatively with the *Odoribacteraceae* family (r = −0.85, *p* = 0.034) (Figure 4C) and *Deltaproteobacteria* class (r = −0.85, *p* = 0.034) (Figure 4D), whereas in 14-week-old BTBR^ob/ob^ FMT (-) and FMT (+) mice, UACR was correlated negatively to the *Lactobacillales* order (r = −0.9, *p* = 0.02 and r = −0.89, *p* = 0.04, respectively) (Figure 4E).

However, BTBR^ob/ob^ mice of all ages, regardless of treatment, exhibited an increase in the accumulation of the mesangial matrix (Figure 4F), a hallmark not only of early stages of DKD but also of progression [1]. In line with these findings, we verified an induction of several genes within the kidneys of BTBR^ob/ob^ mice of all ages, indicating renal hypertrophy and fibrosis E-cadherin, actin alpha 2 (Acta 2), cluster of differentiation (CD90), transforming growth factor beta (TGF-β), sodium-glucose co-transporter 1 (SGLT-1) and sodium-glucose co-transporter 2 (SGLT-2) (Appendix A). Platelet-derived growth factor (PDGF) expression was not upregulated in treated animals, suggesting a potential reduction in mesangial proliferation (Figure 4G).

Plasma evaluation of systemic inflammatory markers, including tumor necrosis factor-α (TNF-α), interleukin-6 (IL-6), and monocyte chemoattractant protein-1 (MCP-1), was similar among mice in all groups (*p* > 0.05) (Figure 4H–J), which points out that FMT was a safe treatment in these animals. TNF-α gene expression was similar in all BTBR^ob/ob^ mice, regardless of time and treatment (*p* > 0.05) (Appendix A).

### 2.5. Evaluation of the Number of Podocytes, Cell Death, and Renal Oxidative Stress in Renal Tissue

To verify the impact of FMT on halting the progression of DKD, we sought to analyze the number of Wilms´Tumour-1 (WT1^+^) podocyte/glomerulus. Podocytopenia was found in all BTBR^ob/ob^ mice, independently of age, and was not prevented by FMT (Figure 5A). Likewise, FMT did not prevent the downregulation of podocyte genes, such as nephrin, podocin, and integrin β1 (Appendix A). Nonetheless, WT-1 expression was maintained after FMT, as opposed to untreated animals, highlighting the potential for preservation of both the podocytes and glomerular filtration barrier (Figure 5B).

We also investigated the 4- hydroxy-2-noneal (4-HNE) lipid peroxidation marker as a result of oxidative stress signaling in the kidney induced by hyperglycemia. Lipid-related oxidative stress was not significantly different in all mice, regardless of age and treatment (*p* > 0.05) (Figure 5C). When we evaluated total caspase-mediated cell death within the kidneys of BTBR WT and BTBR^ob/ob^ mice, no differences were observed in FMT-treated and non-treated animals (Figure 5D). Likewise, cleaved caspase 3 expression did not change in accordance with time and treatment (Figure 5E).

### 2.6. TNF-α Gene Expression and Morphological Parameters in the Ileum and Ascending Colon

FMT decreased TNF-α gene expression in the ileum and preserved the height of the crypt and villi in this segment of the intestine (Figure 6A,B). Similar findings were found in the ascending colon (Figure 6C,D). These findings highlight the safety of FMT and the association of decreased inflammation in the intestine.

Likewise, we did not find significant differences in claudin-1 expression between BTBR WT and BTBR^ob/ob^ mice regardless of time and treatment in the ileum crypt and villus (*p* > 0.05) (Figure 7A) and in the ascending colon crypts (Figure 7B), corroborating the gene expression data. Occludin expression in the ileum crypts and villi and in the ascending colon crypts was similar regardless of age and treatment (Figure 7C,D). Decreased occludin gene expression was prevented by FMT in the ileum and ascending colon of treated mice, but not claudin, zonula occludens-1, and leucine-rich repeat-containing G-protein coupled receptor 5 (LgR5) (Appendix A).

## 3. Discussion

The present study sought to investigate the use of FMT in BTBR^ob/ob^ in mice as a possible therapeutic strategy to mitigate hyperglycemia-driven DKD progression. Our main results showed that FMT is a safe treatment that prevented body weight gain, reduced albuminuria, decreased local expression of TNF-α in the ileum and ascending colon, and potentially ameliorated insulin resistance in BTBR^ob/ob^ mice. To our knowledge, this is the first study using FMT in BTBR^ob/ob^ mice.

Murine models, especially those genetically modified, have been widely used in research on microbiota and FMT, allowing a better understanding of host–microbiota crosstalk [5,16,23,24]. Importantly, FMT efficacy is challenged by several factors, such as route of delivery, amount of feces per sample, number of transplants, condition burden, and target impact [18].

As previously reported, leptin-deficient BTBR^ob/ob^ mutant mice are a hyperphagic model that presents significant body weight gain at an early age [25,26]. Leptin is mainly expressed in adipocytes and has pleiotropic effects in the regulation of energy homeostasis, neuroendocrine function, and immune response, so leptin pathway signaling disruption may lead to metabolic disorders [27,28]. Leptin levels are proportional to body adiposity and are notably elevated in obese individuals [29]. Thus, hyperleptinemia secondary to leptin resistance is implicated in a chronic inflammatory state and ultimately in weight gain [30]. Obese individuals have decreased levels of GLP-1, probably due to leptin resistance [9,31]. However, the reduction in body weight in our treated BTBR^ob/ob^ mice could not be explained either by GLP-1 secretion or by food intake. In a less severe mouse model of leptin deficiency, the burden of obesity and DM is related to a gut microbiota signature, in particular a reduction in a time-dependent manner of *Akkermansia muciniphila* abundance [23]. Additionally, disturbances in the intestinal microbial ecosystem may be associated with low-grade inflammation induced by systemic absorption of lipopolysaccharides derived from the outer membrane of Gram-negative bacteria caused by changes in intestinal permeability [10], which contributes to the progression of obesity and T2DM [32]. Previous reports found a predominance of the *Firmicutes/Bacteroidetes* ratio in obese mice [28,33] and humans [34,35], so that gut microbiome associated with obesity had a higher capacity for energy harvest from the diet [8]. Therefore, that ratio is frequently mentioned as a hallmark of obesity [36]. In our study, most of the sequences obtained by intestinal microbiota analysis revealed greater proportions of the *Bacteroidetes* in comparison to the *Firmicutes* phylum in either obese or non-obese mice, which is in agreement with other human studies [11,37]. However, the literature is poor on gut microbiota characterization in BTBR strains, and most studies addressing gut microbiota composition included other mice strains [8,13,23,28,33]. Additionally, *Firmicutes/Bacteroidetes* ratio involvement in the metabolic parameters is controversial in the literature [11,36,37]. Most studies did not perform longitudinal analysis, including our study, which is a limitation of the interpretation of a gut microbiota signature. Composition and diversity of the gut microbiota is influenced not only by host-dependent and environmental factors [9] but also by leptin deficiency [38]. In treated BTBR^ob/ob^ mice, body weight correlated positively to the *Betaproteobacteria* class, Gram-negative bacteria belonging to the *Proteobacteria* phylum, which are highly enriched in diabetic individuals and positively correlated to plasma glucose [11]. In untreated BTBR^ob/ob^ mice, the abundance of the Gram-negative *Odoribacteraceae* family was reduced when compared to treated mice. That bacteria family is associated with succinate consumption [35]. Importantly, succinate is an intermediate synthetized in the tricarboxylic acid cycle and has been identified as a potential mitochondrial DAMP (damage-associated molecular pattern), which mediates the innate immune response and is implicated in various inflammatory diseases [39]. Additionally, high circulating levels of succinate secondary to higher relative abundance of succinate-producing bacteria (*Prevotellaceae* and *Veillonellaceae*) and lower relative abundance of succinate-consuming bacteria (*Odoribacteraceae* and *Clostridaceae*) are associated with obesity and impaired glucose metabolism [35], which may explain the lower body weight gain in treated BTBR^ob/ob^ mice. A prospective cross-sectional study found a positive association between circulating succinate levels and body mass index (BMI), insulin, glucose, and HOMA-IR in obese individuals, pointing out the influence of microbiota substrates on metabolic profile [35]. Although we did not evaluate endogenous succinate levels, a previous study documented elevated circulating succinate levels in ob/ob mice when compared to healthy controls [40].

Several reports have shown the potential of specific bacterial metabolites as regulators of a range of metabolic functions in the body, whether positive or negative [4,13,15,35,40]. Thus, butyrate is a short-chain fatty acid and is associated with multiple metabolic beneficial effects, including the improvement of insulin resistance [14]. Therefore, modulation of the number of butyrate-producing bacteria could offer protection against T2DM [41]. In our study, the proportion of the two main butyrate-producing bacteria, *Lachnospiraceae,* and *Ruminococcaceae* [14] were similar among treated and untreated mice. In ob/ob mice, gut microbiota abundance of the unclassified genus from the *Lachnospiraceae* family had a negative correlation with the oral glucose tolerance test [14,23]. Male recipients with metabolic syndrome showed improvement in peripheral insulin sensitivity six weeks after receiving allogenic intestinal microbiota from lean donors, attributed to an increase in gut microbial diversity, including those related to butyrate-production [17].

In our study, FMT as a single approach was not sufficient for effective control of glycemic levels. Therefore, the modulation of the intestinal microbiota combined with other established therapies for DM, including lifestyle modifications, pharmacological drugs including metformin [42,43], sodium-glucose cotransporter-2 inhibitors (SGLT2i) [44,45], GLP-1 receptor agonists [46,47], and lipid-lowering drugs, such as statins [48,49], can result in better metabolic parameters, which may ultimately mitigate the damage associated with this complex pathophysiology.

The lack of glycemic control contributed to the maintenance of hypertrophy of pancreatic islets in BTBR^ob/ob^, as described in that model [50]. The compensatory expansion of β-cell mass in response to insulin resistance is a finding in both obese and insulin-resistant animals and humans [51]. However, HOMA-IR and the secretion of insulin and C-peptide in treated mice behaved similarly to BTBR WT mice, suggesting lower insulin resistance after FMT. The incretins GLP-1 and GIP levels, both insulinotropic hormones, were not associated with insulin resistance in our study, although the secretion of these hormones is impaired in obesity and T2DM [52]. Ablation of GIP in Lep^ob/ob^ mice, did not prevent body weight gain and insulin resistance, suggesting that endogenous GIP may not have a role in the development of obesity in leptin deficient mice, indicating that the crosstalk between leptin and GIP secretion warrants further investigation [53].

In our study, FMT prevented the increase in UACR, a time-dependent and reversible feature of BTBR^ob/ob^ mice [19,26], yet the number of podocytes decreased, and the mesangial matrix continued to expand over time. However, gene expression of WT-1 and PDGF indicated a potential reduction in the progression of DKD. Additionally, the slight hyperfiltration observed in untreated mice, yet not significant, could at least in part explain the higher levels of UACR in these animals. The improvement in albuminuria in treated mice may be associated with a hemodynamic mechanism in the kidney, indicating a reduction in glomerular hypertension. It is noteworthy that hyperinsulinemia is involved in increased glomerular hydrostatic pressure and renal vascular permeability, which aggravate glomerular hyperfiltration [54,55]. Thus, FMT promoted a decrease in hyperinsulinemia and insulin resistance and could contribute to preserving the glomerular filtration barrier. In treated mice, we found that albuminuria correlated negatively with the *Odoribacteraceae* family. Although succinate correlated with some metabolic parameters in other studies [35,56,57], its effect on albuminuria levels remains elusive. Albuminuria also correlated negatively with *Deltaproteobacteria* class, Gram-negative bacteria belonging to the *Proteobacteria* phylum [11]. That class contains sulfate-reducing bacteria, which conduct dissimilatory sulfate reduction to acquire energy [58] and produce hydrogen sulfide as a final product [59]. Despite hydrogen sulfide contributing to dysbiosis [60], that metabolite holds several biological properties, such as anti-inflammatory potential and renoprotective effects [61] by modulating the antioxidant response and oxidative stress in the kidney [62]. However, FMT did not affect oxidative stress and the apoptosis-mediated cell death signaling pathway in the kidney, which can be explained by the burden of the DKD microenvironment, as no treatment for DM was performed in our animals.

Importantly, FMT was a safe procedure, and the intestinal architecture, including the height of the villi in the ileum, the depth of the crypts in the ileum or ascending colon, and the expression of tight junction proteins, was not compromised by this treatment. DM adversely impacts these structures and is associated with morphological and physiological remodeling, which leads to the proliferation of the villi and crypt in the ileum, and increased thickness of the ascending colon [63].

Notably, intestinal homeostasis is maintained by the dynamic interaction between luminal microorganisms and the intestinal epithelium [64], so hyperglycemia may predispose individuals to intestinal barrier dysfunction [65]. Changes in the intestinal microbiota are also associated with an increase in intestinal permeability in metabolic disease [66]. Obese mice showed an alteration in the intestinal barrier characterized by the rupture of the tight junction occludin and zonula occludens-1 proteins and indicated that the increase in TNF-α orchestrates this structural damage [21]. Our findings pointed out a reduction in the expression of the inflammatory marker TNF-α in the ileum and ascending colon segments of treated animals, which corroborates FMT safety.

Collectively, our data documented that FMT is a safe treatment and was implicated in albuminuria reduction and decreasing local inflammation in the ileum and ascending colon, and with a trend in ameliorating insulin resistance in BTBR^ob/ob^ mice. These results suggest potentially important beneficial effects of FMT and support further investigation in diabetic patients.

## 4. Research Design and Methods

Experiments were carried out in accordance with the Institutional Animal Care and Use Committees of Hospital Israelita Albert Einstein (HIAE), and the study was registered on the Jewish Institute of Research and Education, HIAE, São Paulo, SP, Brazil (No. 2704-16).

BTBR^ob/ob^ mice (BTBR.Cg-Lep^ob^/WiscJ; #004824-JAX Laboratories) homozygous for the leptin gene knockout is a reversible model for T2DM and DKD, characterized, in males, by early development of obesity, hyperphagia, insulin resistance, and hyperglycemia (6th week of age), followed by DKD establishment with time-dependent albuminuria (8th week of age), resembling human DKD alterations [19]. Male wild-type mice were used as the healthy control and were compared with BTBR^ob/ob^ at different ages (9–11 and 14–15-week-old). Six animals were used in all groups. Mice were housed in a controlled environment (12-h daylight cycle; lights off at 6:00 p.m.) alone or in groups of two or three per cage, with free access to chow (Nuvilab CR-1, Quimtia S/A, Colombo, PR, Brazil) and water *ad libitum*.

At the end of protocol animals (age 14–15 weeks), blood samples from the portal vein and vena cava were collected and stored at −20 °C, whereas kidney, pancreas, ileum, and ascending colon were harvested and stored at −80 °C for molecular analyses.

All experimental protocols were conducted in accordance with the guidelines and regulations of the Association for Assessment and Accreditation of Laboratory Animal Care (AAALAC).

### 4.1. Fecal Microbiota Transplant (FMT)

For FMT, 300 mg of feces from all segments of intestine were collected per sample from BTBR wild-type donors and dissolved in 500 μL phosphate buffered saline (PBS, ThermoFisher Scientific, Waltham, MA, USA). The samples were homogenized using a vortex and centrifuged for 5 min at 6000 rpm at 4 °C to separate the particulate material. FMT was performed on male BTBR^ob/ob^ mice aged 9–11 weeks, referred to as FMT (+). We used the sample supernatant (~300 µL), which was administered via rectal route using a polyethylene probe into the intestine. This probe was introduced approximately 3–3.5 cm. After performing the procedure, the animal was placed in a ventral position with the head down at an angle of 45° for 2–3 min to avoid extravasation of the transplanted material. Then, treated mice were housed individually in cages. Euthanasia was carried out after 4–5 weeks.

### 4.2. Functional Assessment

All mice were weighed once weekly. Urine was obtained at baseline, weeks 9–10, and 13–14 in all groups using a metabolic cage. Albuminuria levels were measured by an Albumin Mouse ELISA Kit (Abcam, ab207620, Cambridge, MA, USA) and were standardized to urinary creatinine, assessed by a colorimetric assay Creatinine K^®^ 96–300 (Labtest, Vista Alegre, Brazil) using a biochemical analyzer Cobas Mira Plus (Roche, Basel, Switzerland). The results were reported as urinary albumin-to-creatinine ratio (UACR; g/mg). Blood glucose was quantified with a glucometer (Accu-Check Advantage Blood Glucose Monitor^®^ (Roche Diagnostic Corporation, Indianapolis, IN, USA) after six hours of fasting. All procedures were performed on animals at the beginning (9–11 weeks-old) and at the end of the protocol (14–15 weeks-old). Glycosuria was measured by the Glucose Liquiform^®^ 133-2/500 LabTest kit (Lagoa Santa, MG, Brazil), and the absorbance values were obtained by the ChemWell-T spectrophotometer LabTest (Lagoa Santa, MG, Brazil). Homeostasis Model Assessment-Insulin Resistance (HOMA-IR) and HOMA-β, as previously described [67].

### 4.3. Microbial Analysis of the Intestinal Contents of Mice

After euthanasia, fecal content was collected from all segments of the intestine and analyzed using a QIAamp DNA Stool Mini Kit^®^ (Qiagen, Hilden, Germany).

### 4.4. Evaluation of Metabolic Parameters and Inflammatory Markers

Blood was collected from the portal vein, and we immediately added 10 µL of DPP4 inhibitor (DPP IV Inhibitor^®^, Millipore, Burlington, MA, USA) and 10 µL of aprotinin (Aprotinin, Bovine Lung, Crystalline^®^, Millipore, Burlington, MA, USA) for glucagon-like peptide-1 (GLP-1) and glucagon analyses, respectively. GLP-1, glucose-dependent insulin tropic polypeptide (GIP), peptide YY (PYY), and plasma cytokines tumor necrosis factor (TNFα), interleukin (IL)-6, monocyte chemoattractant protein (MCP-1), insulin, C-peptide, and glucagon levels were determined in duplicate using the kit MILLIPLEX MAP Mouse Metabolic Hormone Magnetic Bead Panel-Metabolism Multiplex Assay^®^ (Millipore, Bellerica, MA, USA), and measured by using Luminex technology (Bio-Rad Bioplex). To quantify GLP-2, we used the KIT ELISA EZGLP2-37K (Millipore, Bellerica, MA, USA).

### 4.5. Mesangial Expansion

Kidney sections (paraffin-fixed, 3–4 µm thick) were stained with periodic acid-Schiff reagent (PAS; Sigma-Aldrich, St. Louis, MO, USA). The increase in the mesangial matrix was measured by the presence of PAS-positive area in the mesangium and was defined by percentage using light microscopy (magnification 40×; Olympus). The glomerular area (μm^2^) was also indicated along the outline of capillary loops using CellSens software (Olympus) in 30 randomly selected glomeruli in each animal.

### 4.6. Morphological Analysis

Pancreas sections (paraffin-fixed, 3–4 µm thick) were stained with hematoxylin-eosin (HE) to assess the number, volume, and integrity of pancreatic islets using light microscopy (magnification 10×). We counted six islets on average per animal.

Ileum and ascending colon sections (paraffin-fixed, 3–4 µm thick) were stained with HE to access villi and crypts using light microscopy (magnification 20×). In 10 villi per sample, we measured the height of the villi in the ileum, as well as in 10 crypts in the ileum and ascending colon.

### 4.7. Immunohistochemistry (IHC) Analyses

BTBR^ob/ob^ and wild-type mice tissues (kidney, ileum, and ascending colon) were preserved with formalin (10%) and sectioned at 3–4 µm thick. Quantification analyses were carried out by CellSens software (Olympus) using 20× magnification.

For kidney analyses, the score for WT-1^+^ cells was performed by counting the number of positive nuclei in 25 randomly selected glomeruli in the kidney cortex using 100× magnification and after applying rabbit polyclonal anti-WT-1 (Santa Cruz, Dallas, TX, USA). All data were pooled to obtain the number of WT-1^+^ podocytes per glomerular cross-section. For apoptosis pathway analysis, we evaluated caspase-3 and cleaved caspase-3 protein activation (#9662 and 9661, Cell Signaling, Danvers, MA, USA), and for oxidative stress, we estimated the 4-HNE detection (anti-4 hydroxynonenal rabbit polyclonal, ab46545, Abcam, Cambridge, MA, USA). IHC reaction was carried out using the EnVision FLEX High pH kit (K8000, DAKO, Santa Clara, CA, USA).

For ileum and ascending colon analyses, we verified the expression of tight junction claudin and occludin (anti-claudin 1 rabbit polyclonal, ab15098, Abcam, Waltham, MA, USA, and anti-occludin rabbit polyclonal, 40-4700, ThermoFisher Scientific, Waltham, MA, USA). See Appendix A for detailed procedure. The sections were initially blocked with a Vector SP 2001^®^ kit (Vector, Burlingname, CA, USA). For blocking unspecific connections, 1:20 goat non-immune serum (Sigma-Aldrich, Burlington, MA, USA) was used for 30 min, followed by incubation with primary antibodies overnight in a humid chamber at 4 °C. Slides were then washed in TBS-T buffer and incubated for 45 min with a secondary goat anti-rabbit antibody bound to biotin. The slides were washed again with TBS-T and incubated with avidin-biotin-HRP complex (Vector AK 4000, Burlingame, CA, USA) for 30 min. DAB staining was used for a maximum of 10 min. Tissue sections were then counterstained with Mayer’s hemalum. Three to ten villi per field were evaluated to measure the height of the crypts in the ileum and ascending colon and the height of the villi in the ileum.

### 4.8. RNA Extraction and cDNA Synthesis

RNA from tissues (kidney, ileum, and ascending colon) was prepared using RNeasy^®^ Fibrous Tissue Mini Kit (Qiagen, Hilden, Germany) kit. cDNA was synthesized using the High-capacity cDNA Reverse Transcription Kit (ThermoFisher Scientific, Waltham, MA, USA).

### 4.9. Real-Time qPCR Assays

qPCR reactions were performed on the Thermocycler qPCR-QuantStudio 6 Flex System using the TaqMan Real-Time PCR Master Mix kit (ThermoFisher Scientific, Waltham, MA, USA).

In the kidney, we used the following probes: β1-integrin (Mm01253230-m1), CD90 (Th1: Mm00493682_g1), Podocin (Nth2: Mm01292252-m1), α-Actin (Acta2: Mm00725412_s1), E-cadherin (Cdh1: Mm01247357_m1), Nephrin (Nph1s: Mm01176615_g1), Wilms tumor protein (Wt1: Mm01337048_m1), Tumor Necrosis Factor-α (TNF-α: Mm00443258_m1), Transforming Growth Factor-ß (TGF-β; Mm01178820), Platelet-derived Growth Factor (PDGF; Mm00440677), Sodium Glucose Transporter-1 (SGLT-1; Mm00451203), and SGLT-2 (Mm00453831). In the ileum and ascending colon, TaqMan probes were used to target the following genes: Claudin 3 (Mm00515499), Occludin (Mm00500912), Zonula Ocludens-1 (Mm00493699), Leucine-rich repeat containing G-protein coupled receptor 5 (LGR5; Mm00438890), and Tumor Necrosis Factor-α (TNF-α: Mm00443258_m1). TaqMan GAPDH (Mm99999915_g1) and GUSB: (Mm01197698_m1) probes were used as endogenous controls. All rigs were purchased from Thermo Fisher Scientific–Waltham, MA, USA. We analyzed the Ct (“Cycle threshold”) values using QuantStudio6 Flex System Software. To determine the relative expression of the values, method 2–ΔΔCt (fold change) was used, where averages of duplicates of the Ct values were calculated for each sample and subtracted from those derived from the GUSB.

### 4.10. Preparation of Sequencing Libraries for Microbial Analysis of Intestinal Content of Mice

The preparation of the sequencing libraries was carried out in a two-step PCR protocol, using primers for V3–V4 regions as previously described. PCR reactions were performed in triplicate. The final PCR reaction was cleaned using AMPurebeads (Beckman Coulter, Brea, CA, USA), and the samples were pooled in the sequencing libraries for quantification. Pool amplification estimates were performed with Picogreen dsDNA assays (Invitrogen, Waltham, MA, USA) and then the pooled libraries were diluted for quantification via qPCR using the KAPA Library Quantification Kit for Illumina platforms (KAPA Biosystems, Woburn, MA, USA). The libraries were sequenced on a MiSeq system using the standard Illumina primers provided in the kit. After sequencing, the bioinformatics pipeline performs demultiplexing of sequences, removal of adapters, and trimming of primers. The readings were normalized in size of 283 bp, and other analyses were performed in the statistical analysis environment R.

Data manipulation was performed using the packages tidy verse (version 1.2.1) and phyloseq (summer 1.28.0). The ggplot2 package (version 3.2.0) was used for data visualization. Differential abundance analysis was performed using the DESeq2 package (version 1.2.2). The method assumes that the observed counts follow a negative binomial distribution, which is analyzed using a generalized linear model. Regression coefficients were analyzed using the Wald test. *p*-values were corrected for multiple comparisons using the Benjamin Hochberg procedure, controlling the rate of false discoveries by 10%. Nonparametric analysis, when necessary, included Kruskal–Wallis and Wilcoxon tests.

### 4.11. Statistical Analysis

The results were expressed as the mean ± SEM (standard error of the mean) or median and interquartile range (IQR). Statistical analysis was performed using the Shapiro–Wilk normality test. ANOVA (Analysis of Variance) was performed in samples with normal distribution, followed by Tukey’s post-test. For repeated measures over time, a two-away ANOVA/mixed model test was performed (when there were differences in the number of observations in repeated measures). For multiplex comparisons when baseline differences were highly variable, the Fisher LSD post-test was performed. In case of non-normal distribution of the sample means, we used the Kruskal–Wallis or t test with Holm–Sidak correction for the multiple comparisons. Evaluation was made using Graph Pad Prism version 8.2.1 for Windows Vista (Graph Pad Software, San Diego, CA, USA). Statistical analysis of the sequencing data for the 16S region of the ribosomal RNA was performed using R software (version 3.6.3). *p* < 0.05 was considered statistically significant.

## Figures and Tables

**Figure 1 ijms-23-03842-f001:**
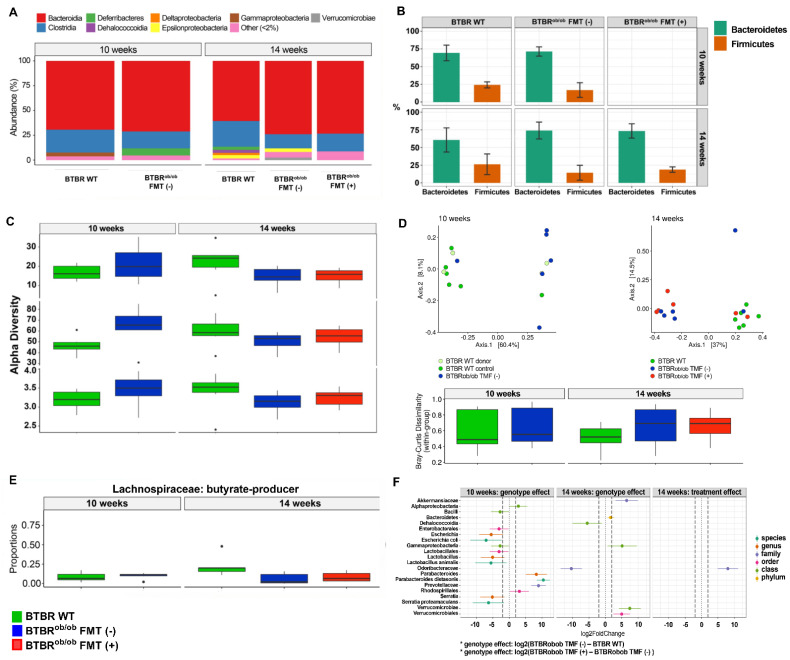
Intestinal composition analysis by 16S rRNA sequencing in BTBR^ob/ob^ FMT (+) and BTBR^ob/ob^ FMT (-) mice compared to BTBR WT mice. (**A**) Bacterial phyla according to 16S rRNA sequencing of 10- and 14-week-old BTBR^ob/ob^ FMT (+) and FMT (-) mice compared to age-matched BTBR WT mice: *Bacteroidetes*, *Firmicutes*, *Proteobacteria*, *Actinobacteria*, *Verrucomicrobia*, *Deferribacteres*, and *Chloroflex.* (**B**) Alpha diversity analysis by Shannon Index in 10-week-old BTBR WT and age-matched BTBR^ob/ob^ mice (*p* = 0.24), and among 14-week-old BTBR WT *versus* BTBR^ob/ob^ FMT (-) and BTBR^ob/ob^ FMT (+) mice (*p* = 0.21) indicated that the species richness was similar among groups. Results are median and IQR. (**C**) Principal coordinate analysis (PCoA) based on Bray–Curtis dissimilarity metrics did not show any clearly associated clusters in relation to the study animals. Results are median and IQR. (**D**) Butyrate producer *Lachnospiraceae* bacteria family proportions were similar between 10-week-old BTBR WT and age-matched BTBR^ob/ob^ mice (*p* = 0.59), yet higher proportions of these bacteria in 14-week-old BTBR WT compared to 14-week-old BTBR^ob/ob^ FMT (-) and FMT (+) mice (* *p* = 0.026) were observed. No significant difference was found between 14-week-old BTBR^ob/ob^ FMT (+) and FMT (-) mice (*p* = 0.39). Results are median and IQR. (**E**,**F**) Assessment of the differences in relative abundance between genotypes effect evaluated by log 2-fold change demonstrated greater relative abundance in the *Gammaproteobacteria* and *Verrucomicrobiae* classes and lower abundance in the *Dehalococcoidia* and *Odoribacteraceae* families in 14-week-old BTBR^ob/ob^ FMT (-) mice. The treatment increased the abundance of the *Odoribacteraceae* family in 14-week-old BTBR^ob/ob^ FMT (+) mice. In all analyses, *n* = 6/group.

**Figure 2 ijms-23-03842-f002:**
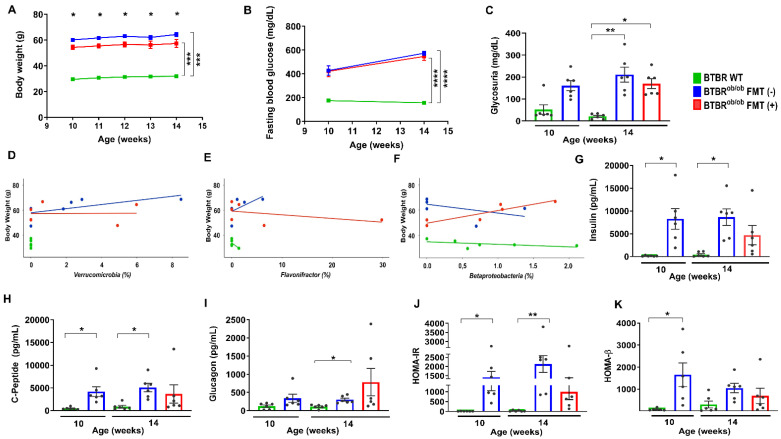
Analysis of the functional and metabolic parameters in BTBR^ob/ob^ FMT (+) and BTBR^ob/ob^ FMT (-) mice compared to BTBR WT mice. (**A**) BTBR^ob/ob^ mice showed significantly higher body weight when compared to BTBR WT mice (*** *p* = 0.0001); however, 14-week-old BTBR^ob/ob^ FMT (+) mice at all intervals gained less body weight than BTBR^ob/ob^ FMT (-) mice (Row 1: * *p* = 0.03, Row 2: * *p* = 0.02, Row 3: * *p* = 0.02, Row 4: * *p* = 0.04, Row 5: * *p* = 0.01). (**B**) Fasting blood glucose was higher in 10-week-old BTBR^ob/ob^ mice when compared to age-matched BTBR WT mice (*p* = 0.0001), and between 14-week-old BTBR^ob/ob^ FMT (-) and FMT (+) mice when compared to age-matched BTBR WT mice (**** *p* < 0.0001). No difference was found between 14-week-old BTBR^ob/ob^ FMT (-) and FMT (+) mice (*p* = 0.78). (**C**) Glycosuria did not change significantly between 10-week-old BTBR WT and age-matched BTBR^ob/ob^ mice (*p* = 0.59). However, it was higher between 14-week-old BTBR WT and 14-week-old BTBR^ob/ob^ FMT (-) (** *p* = 0.002) and FMT (+) (** *p* = 0.02) mice, yet with no significant difference between 14-week-old BTBR^ob/ob^ FMT (-) and FMT (+) mice (*p* > 0.99). (**D**,**E**) The body weight of 14-week-old BTBR^ob/ob^ FMT (-) mice correlated positively to the *Verrucomicrobia* phylum (r = 0.9; * *p* = 0.01) (**D**) and *Flavonifractor* genus (r *=* 0.88 * *p =* 0.02) (**E**). (**F**) Body weight of 14-week-old BTBR^ob/ob^ FMT (+) mice was positively associated with the intestinal *Betaproteobacteria* class (r = 0.93, ** *p* = 0.008). (**G**) Plasma insulin was elevated in 10-week-old BTBR^ob/ob^ when compared to age-matched BTBR WT mice (* *p* = 0.01) and between 14-week-old BTBR WT mice *versus* BTBR^ob/ob^ FMT (-) mice (* *p* = 0.02). No significant difference was found between 14-week-old BTBR WT and BTBR^ob/ob^ FMT (+) mice (*p* = 0.3) and 14-week-old BTBR^ob/ob^ FMT (-) and BTBR^ob/ob^ FMT (+) mice (*p* > 0.99). (**H**) Plasma C-peptide was higher in 10-week-old BTBR^ob/ob^ mice when compared to age-matched BTBR WT mice (* *p* = 0.01) and between 14-week-old BTBR WT and BTBR^ob/ob^ FMT (-) mice (* *p* = 0.02). No significant difference was found between 14-week-old BTBR^ob/ob^ FMT (-) and FMT (+) mice (*p* > 0.05) and 14-week-old BTBR WT and BTBR^ob/ob^ FMT (+) mice (*p* = 0.79). (**I**) Plasma glucagon did not change significantly between 10-week-old BTBR WT and age-matched BTBR^ob/ob^ mice (*p* = 0.57). However, glucagon levels were higher in 14-week-old BTBR^ob/ob^ FMT (-) mice when compared to 14-week-old BTBR WT (* *p* = 0.02) but not to BTBR^ob/ob^ FMT (+) mice (*p* > 0.99). There was a trend toward higher glucagon levels in BTBR^ob/ob^ FMT (+) mice in comparison to age-matched BTBR WT mice (*p* = 0.05). (**J**) HOMA-IR in 10-week-old BTBR^ob/ob^ was elevated when compared to age-matched BTBR WT mice (* *p* = 0.02) and between 14-week-old BTBR WT *versus* BTBR^ob/ob^ FMT (-) mice (** *p* = 0.006). However, no difference was observed between 14-week-old BTBR WT and 14-week-old BTBR^ob/ob^ FMT (+) mice (*p* = 0.2) and between 14-week-old BTBR^ob/ob^ FMT (-) and FMT (+) mice (*p* > 0.99). (**K**) HOMA-β was significantly higher in 10-week-old BTBR^ob/ob^ when compared to age-matched BTBR WT mice (*p* = 0.01) and did not change significantly between 14-week-old BTBR WT and BTBR^ob/ob^ FMT (-) and BTBR^ob/ob^ FMT (+) mice (*p* = 0.22 and *p* > 0.99, respectively). All data are means ± SEM. In all analyses, *n* = 6/group.

**Figure 3 ijms-23-03842-f003:**
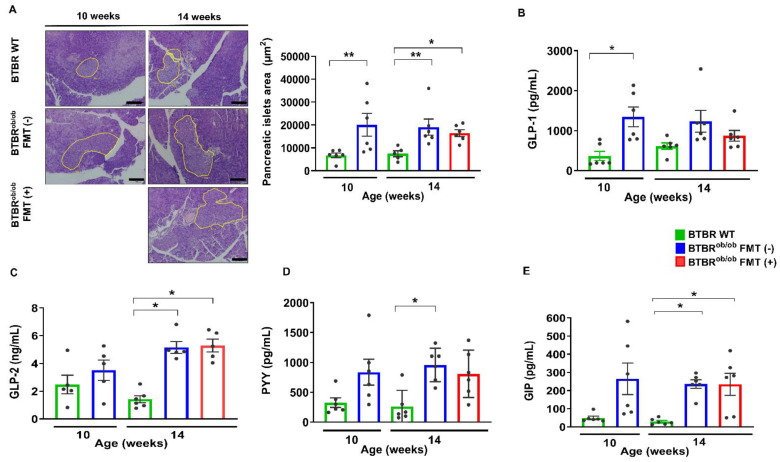
Analysis of morphological and metabolic parameters in BTBR^ob/ob^ FMT (+) and FMT (-) mice compared to BTBR WT mice. (**A**) Representative hematoxylin-eosin (HE) staining of morphological evaluation of pancreatic islets in 10-week-old BTBR WT compared to age-matched BTBR^ob/ob^ mice and 14-week-old BTBR WT compared to BTBR^ob/ob^ FMT (+) and FMT (-) mice. Data exhibited hypertrophy of the pancreatic islet in 10-week-old BTBR^ob/ob^ mice when compared to age-matched BTBR WT mice (** *p* = 0.005), and between BTBR^ob/ob^ FMT (-) (** *p* = 0.008) and FMT (+) (* *p* = 0.01) *versus* 14-week-old BTBR WT mice. No difference was observed between BTBR^ob/ob^ FMT (-) and FMT (+) mice (*p* = 0.93). (**B**) Plasma GLP-1 was significantly more elevated in 10-week-old BTBR^ob/ob^ mice when compared to age-matched BTBR WT mice (* *p* = 0.01) but did not show a significant difference between 14-week-old BTBR WT *versus* 14-week-old BTBR^ob/ob^ FMT (-) (*p* = 0.23) and FMT (+) mice (*p* > 0.99). (**C**) Plasma GLP-2 was significantly elevated in 14-week-old BTBR^ob/ob^ FMT (-) and BTBR^ob/ob^ FMT (+) mice when compared to age-matched BTBR WT mice (* *p* = 0.02 and * *p* = 0.01, respectively), but no difference was found between BTBR^ob/ob^ 14-week-old FMT (-) and FMT (+) mice (*p* > 0.99). (**D**) Plasma PYY was significantly different between 14-week-old BTBR WT and 14-week-old BTBR^ob/ob^ FMT (-) (* *p* = 0.02) mice, although no difference was observed between 14-week-old BTBR WT and BTBR^ob/ob^ FMT (+) mice (*p* > 0.99). (**E**) Plasma GIP was significantly higher in 14-week-old BTBR^ob/ob^ FMT (-) and BTBR^ob/ob^ FMT (+) mice when compared to age-matched BTBR WT mice (* *p* = 0.02 for both), but no difference was found between BTBR^ob/ob^ 14-week-old FMT (-) and BTBR^ob/ob^ FMT (+) mice (*p* > 0.99). All data are means ± SEM. Scale bars represent 100 µm in (**A**). In all analyses, *n* = 5–6/group.

**Figure 4 ijms-23-03842-f004:**
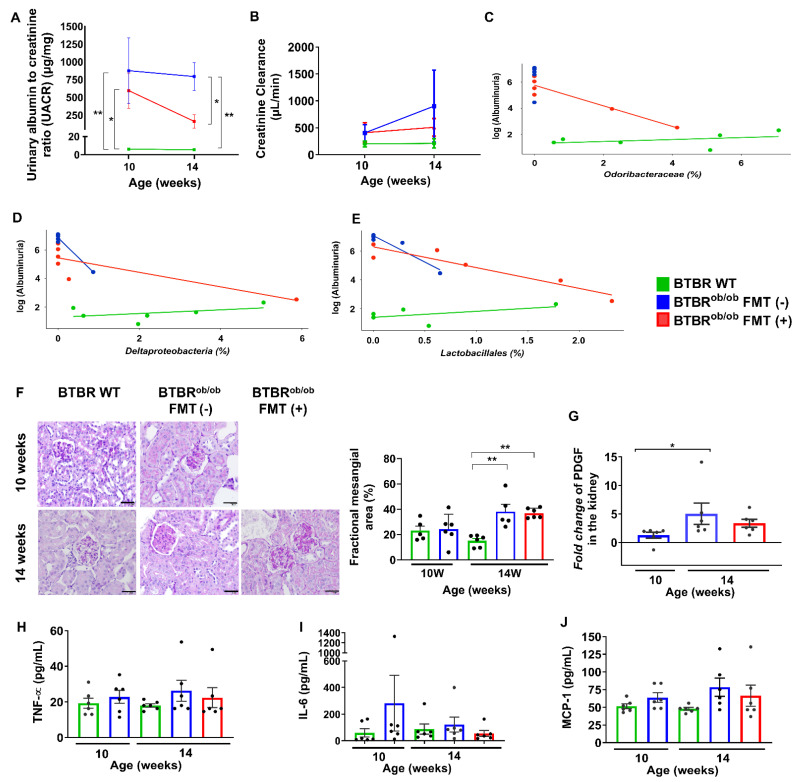
Functional and morphological evaluation of the kidney, metagenomics analysis, and systemic inflammatory markers. (**A**) Urinary albumin-to-creatinine ratio (UACR; μg/mg) was higher in 10-week-old BTBR^ob/ob^ mice when compared to the UACR of age-matched BTBR WT mice (** *p*= 0.006). UACR in 14-week-old BTBR^ob/ob^ FMT (-) mice was significantly more elevated when compared to age-matched BTBR WT mice (** *p* = 0.007). Additionally, UACR in 14-week-old BTBR^ob/ob^ FMT (+) mice was significantly lower when compared to BTBR^ob/ob^ FMT (-) mice (* *p* = 0.03), and no significant difference was found when compared to 14-week-old BTBR WT mice (*p* = 0.55). (**B**) Glomerular filtration rate in accordance with treatment and age: 14-week-old BTBR WT *versus* BTBR^ob/ob^ FMT (-) (*p* = 0.26), 14-week-old BTBR WT *versus* BTBR^ob/ob^ FMT (+) (*p* = 0.63), and BTBR^ob/ob^ FMT (-) *versus* FMT (+) mice (*p*= 0.78). (**C**,**D**) UACR of 14-week-old BTBR^ob/ob^ FMT (+) mice correlated negatively to the *Odoribacteraceae* family (r = −0.85; *p* = 0.034) and *Deltaproteobacteria* class (r = −0.85; *p* = 0.034). (**E**) UACR in 14-week-old BTBR^ob/ob^ FMT (-) and FMT (+) mice interacted negatively with the *Lactobacillales* order (r = −0.9; *p* = 0.02 and r = −0.89; *p* = 0.04 respectively). (**F**) Representative of periodic acid-Schiff (PAS) staining of kidney sections in BTBR WT, BTBR^ob/ob^ FMT (-), and FMT (+) mice between 10 and 14 weeks of age. 14-week-old BTBR^ob/ob^ FMT (-) and FMT (+) mice showed an increase in the accumulation of mesangial matrix when compared to age-matched BTBR WT mice (** *p* = 0.002 for both), and FMT did not prevent that structural damage in 14-week-old BTBR^ob/ob^ compared to untreated mice (*p* = 0.98). (**G**) Fold change of PDGF expression in the kidney in relation to age-matched BTBR WT: 10-week-old BTBR^ob/ob^ mice *versus* 14-week-old BTBR^ob/ob^ FMT (-) mice was significantly different (* *p* = 0.02), but no difference was found in 10-week-old BTBR^ob/ob^ mice *versus* 14-week-old BTBR^ob/ob^ FMT (+) mice (*p* = 0.05), and between 14-week-old BTBR^ob/ob^ FMT (-) and FMT (+) mice (*p* > 0.99). (**H**–**J**) Plasma evaluation of systemic inflammatory markers TNF-α (**H**), IL-6 (**I**), and MCP-1 (**J**) had no significant difference in accordance to the age and treatment (*p* > 0.99). All data are means ± SEM. Scale bars represent 20 µm in (**F**). In all analyses, *n* = 5–6/group.

**Figure 5 ijms-23-03842-f005:**
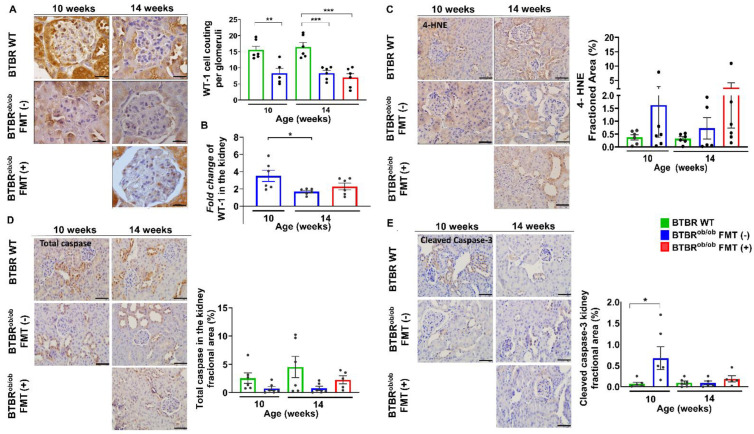
Immunohistochemical analysis of WT-1, total caspase, cleaved caspase-3, and 4-hydroxy-2-noneal (4-HNE) in the kidneys of BTBR^ob/ob^ FMT (-) and FMT (+) mice compared to BTBR WT mice. (**A**) BTBR^ob/ob^ mice exhibited lower detection of WT-1^+^ cells in all ages when compared to BTBR WT mice: 10-week-old BTBR^ob/ob^ and age-matched BTBR WT mice (** *p* = 0.004), 14-week-old BTBR WT *versus* 14-week-old BTBR^ob/ob^ FMT (-) (*** *p* = 0.0007) and FMT (+) (*** *p* = 0.0001) mice. FMT did not prevent a decrease in the podocyte number (*p* = 0.93). (**B**) Fold change of WT-1 expression in the kidney in relation to age-matched BTBR WT: 10-week-old BTBR^ob/ob^ mice *versus* 14-week-old BTBR^ob/ob^ FMT (-) mice was significantly different (* *p* = 0.03), but no difference was found in 10-week-old BTBR^ob/ob^ mice *versus* 14-week-old BTBR^ob/ob^ FMT (+) mice (*p* = 0.16), and between 14-week-old BTBR^ob/ob^ FMT (-) and FMT (+) mice (*p* = 0.62). (**C**) Immunohistochemical analysis for lipid-related oxidative stress was not significantly different between 10-week-old and BTBR^ob/ob^ and age-matched BTBR WT mice (*p* = 0.7), and 14-week-old BTBR WT *versus* 14-week-old BTBR^ob/ob^ FMT (-) mice (*p* = 0.89), 14-week-old BTBR WT *versus* 14-week-old BTBR^ob/ob^ FMT (+) mice (*p* = 0.1), and 14-week-old BTBR^ob/ob^ FMT (-) *versus* 14-week-old BTBR^ob/ob^ FMT (+) mice (*p* = 0.16). (**D**) Immunohistochemical analysis for total caspase showed no significant difference according to age and treatment: 10-week-old BTBR^ob/ob^
*versus* age-matched BTBR WT mice (*p* = 0.7), 14-week-old BTBR WT *versus* 14-week-old BTBR^ob/ob^ FMT (-) mice (*p* = 0.09), 14-week-old BTBR WT *versus* 14-week-old BTBR^ob/ob^ FMT (+) mice (*p* = 0.57), and 14-week-old BTBR^ob/ob^ FMT (-) *versus* 14-week-old BTBR^ob/ob^ FMT (+) mice (*p* = 0.85). (**E**) Immunohistochemical analysis for cleaved caspase-3 showed a significant difference between 10-week-old BTBR^ob/ob^ and age-matched BTBR WT mice (* *p* = 0.03), but no significant difference was found between 14-week-old BTBR WT *versus* 14-week-old BTBR^ob/ob^ FMT (-) mice (*p* > 0.99), 14-week-old BTBR WT *versus* 14-week-old BTBR^ob/ob^ FMT (+) mice (*p* = 0.99), and 14-week-old BTBR^ob/ob^ FMT (-) *versus* 14-week-old BTBR^ob/ob^ FMT (+) mice (*p* = 0.99). Scale bars represent 10 µm in (**A**) and 20 µm in (**C**–**E**). All data are means ± SEM. In all analyses, *n* = 5–6/group.

**Figure 6 ijms-23-03842-f006:**
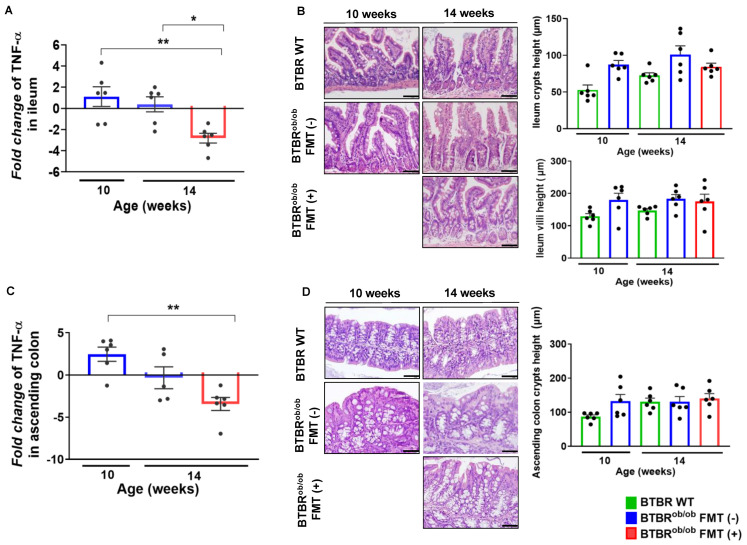
Evaluation of gene expression and morphological aspects of intestinal crypts and villi in BTBR^ob/ob^ FMT (-) and BTBR^ob/ob^ FMT (+) mice compared to BTBR WT mice. (**A**) Fold change of TNF-α expression in the ileum from age-matched BTBR WT: TNF-α expression of 14-week-old BTBR^ob/ob^ FMT (+) mice was significantly downregulated when compared to 10-week-old BTBR^ob/ob^ mice (** *p* = 0.004) and 14-week-old BTBR^ob/ob^ FMT (-) mice (* *p* = 0.02). No significant difference was found between 10-week-old BTBR^ob/ob^
*versus* 14-week-old BTBR^ob/ob^ FMT (-) mice (*p* = 0.76). (**B**) Representative hematoxylin-eosin (HE) staining of the morphological evaluation of ileum crypts from 10-week-old BTBR^ob/ob^ compared to age-matched BTBR WT showed no difference in crypt height (*p* = 0.08), also from 14-week-old BTBR WT *versus* 14-week-old BTBR^ob/ob^ FMT (-) (*p* = 0.49) and FMT (+) (*p* > 0.99) mice. No difference was observed between 14-week-old BTBR^ob/ob^ FMT (-) and FMT (+) mice (*p* > 0.99). Furthermore, no difference was found in the villi of this segment in 10-week-old BTBR^ob/ob^ mice and age-matched BTBR WT mice (*p* = 0.16), and between 14-week-old BTBR WT and 14-week-old BTBR^ob/ob^ mice FMT (-) (*p* > 0.99) and FMT (+) (*p* > 0.99) mice. No difference was observed between 14-week-old BTBR^ob/ob^ FMT (-) and FMT (+) mice (*p* = 0.99). (**C**) Fold change of TNF-α expression in the ascending colon relative to age-matched BTBR WT: TNF-α expression of 14-week-old BTBR^ob/ob^ FMT (+) mice was significantly downregulated when compared to 10-week-old BTBR^ob/ob^ mice (** *p* = 0.0014), but was not significantly different from 14-week-old BTBR^ob/ob^ FMT (-) mice (*p* = 0.1). (**D**) Representative hematoxylin-eosin (HE) staining from morphological assessment of ascending colon crypts in 10-week-old BTBR^ob/ob^ mice and age-matched BTBR WT and 14-week-old BTBR WT mice in comparison to BTBR^ob/ob^ FMT (-) and FMT (+) mice. The data showed no difference in the height of 10-week-old BTBR^ob/ob^ crypts when compared to age-matched BTBR WT (*p* = 0.18) mice and between 14-week-old BTBR WT and 14-week-old BTBR^ob/ob^ FMT (-) (*p* > 0.99) and FMT (+) (*p* = 0.99) mice. There was no difference between 14-week-old BTBR^ob/ob^ FMT (-) and FMT (+) mice (*p* = 0.99). All data are means ± SEM. Scale bars represent 50 µm in (**B**) and 20 µm in (**D**). In all analyses, *n* = 5–6/group.

**Figure 7 ijms-23-03842-f007:**
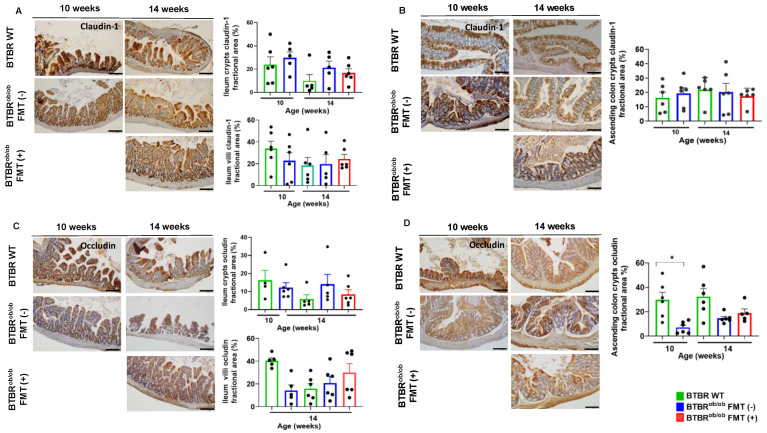
Immunohistochemical evaluation of claudin-1 and occludin proteins in ileum and ascending colon in BTBR^ob/ob^ FMT (+) and BTBR^ob/ob^ FMT (-) mice compared to BTBR WT mice. (**A**) Claudin-1 evaluation in the ileum crypts showed no significant difference according to age and treatment (all *p* > 0.99). Likewise, detection of claudin-1 in ileum villi was not statistically significant: 14-week-old BTBR WT *versus* 14-week-old BTBR^ob/ob^ FMT (-) (*p* > 0.99) and BTBR^ob/ob^ FMT (+) (*p* = 0.97) mice, and 14-week-old BTBR^ob/ob^ FMT (-) *versus* FMT (+) (*p* = 0.99) mice. (**B**) Claudin-1 expression in ascending colon crypts was not different among mice, regardless of age and treatment (all *p* > 0.99). (**C**) Occludin expression in ileum crypts was not significantly different between 14-week-old BTBR WT *versus* 14-week-old BTBR^ob/ob^ FMT (-) (*p* = 0.79) and FMT (+) (*p* > 0.99) mice, and 14-week-old BTBR^ob/ob^ FMT (-) *versus* FMT (+) (*p* > 0.99) mice. In ileum villi, occludin expression was not different between 14-week-old BTBR WT *versus* 14-week-old BTBR^ob/ob^ FMT(-) and FMT (+) (*p* > 0.99 for both) mice, and 14-week-old BTBR^ob/ob^ FMT(-) *versus* FMT (+) (*p* > 0.99) mice. (**D**) Occludin expression in the ascending colon crypt was significantly different between 10-week-old BTBR^ob/ob^ and age-matched BTBR WT mice (* *p* = 0.01), but not significantly different between 14-week-old BTBR WT *versus* 14-week-old BTBR^ob/ob^ FMT (-) (*p* = 0.06) and FMT (+) (*p* = 0.27) mice, and 14-week-old BTBR^ob/ob^ FMT (-) *versus* FMT (+) (*p* = 0.96) mice. All data are means ± SEM. Scale bars represent 100 µm in (**A**–**D**). All data are means ± SEM. In all analyses, *n* = 4–6/group.

## Data Availability

The datasets generated during and/or analyzed during the current study are available from the corresponding author upon reasonable request.

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
