# Peer review of "Fecal Microbiota Transplant in a Pre-Clinical Model of Type 2 Diabetes Mellitus, Obesity and Diabetic Kidney Disease"

_ijms, 2022, doi:10.3390/ijms23073842_

Round 1

Reviewer 1 Report

The manuscript has been modified according to the suggestions 

Author Response

Thank you for supporting our study. 

Reviewer 2 Report

Reviewer comments and suggestions

The research article suggested the role of modulation of the intestinal microbiota in diabetes manifestations and concerning the immunomodulatory function of fecal microbiota transplantation (FMT) in experimental models. The authors evidenced albuminuria reduction and decreasing local inflammation in the ileum and ascending colon, and with a trend in ameliorating insulin resistance in BTBR ob/ob mice. The morphological evaluation by using staining techniques enhances the impact of and the importance of the study in in-vivo methods. In totally, the study suggested the safety and effectiveness of FMT in a preclinical model of type 2 DM, obesity and DKD.

Few suggestions for improvement in the current form of the manuscript as below: Results from immunohistochemical analysis particularly of kidney sections are quite impressive and self-explanatory. But I would highly encourage the photo quality or the image capturing with the same background settings as in Figure No. 6 b, the background of the image is not the same with all the groups that are giving the staining difference in the groups, which might not be the case, Please specify.

The paper has nicely written. However, in many places, the authors need to modify and there were a few inconsistencies in the manuscript for example “The gut microbiota is a complex microbial community that plays an important role in many aspects of human health [4]. . .” Here double full stop, which needs to be deleted.

Page number 12 in discussion, why these lines were italic

In our study, FMT as monotherapy, was not effective in glycemic control, which may be explained by the fact that a single approach may not be sufficient for such effective control.

“That class” why it was italic

Reference: line 50 check the journal format

Reference 68 Not important to cite this paper as this was not a journal. Better to add another reference for this

Author Response

We would like to thank the reviewer for the suggestions. We made the corrections accordingly. They are highlighted in yellow throughout the manuscript.  

The research article suggested the role of modulation of the intestinal microbiota in diabetes manifestations and concerning the immunomodulatory function of fecal microbiota transplantation (FMT) in experimental models. The authors evidenced albuminuria reduction and decreasing local inflammation in the ileum and ascending colon, and with a trend in ameliorating insulin resistance in BTBR ob/ob mice. The morphological evaluation by using staining techniques enhances the impact of and the importance of the study in in-vivo methods. In totally, the study suggested the safety and effectiveness of FMT in a preclinical model of type 2 DM, obesity and DKD.  

1) Few suggestions for improvement in the current form of the manuscript as below: Results from immunohistochemical analysis particularly of kidney sections are quite impressive and self-explanatory. But I would highly encourage the photo quality or the image capturing with the same background settings as in Figure No. 6 b, the background of the image is not the same with all the groups that are giving the staining difference in the groups, which might not be the case, Please specify.

Response: Thank you for your suggestion. We upload new images of the same animals and standardized the background in Figure 6.

2) The paper has nicely written. However, in many places, the authors need to modify and there were a few inconsistencies in the manuscript for example “The gut microbiota is a complex microbial community that plays an important role in many aspects of human health [4]. . .” Here double full stop, which needs to be deleted.

Response: Thank you for your comment. We reviewed the manuscript accordingly.  

3) Page number 12 in discussion, why these lines were italic

Response: Thank you for pointing this out. We corrected the unnecessary italic terms.

4) In our study, FMT as monotherapy, was not effective in glycemic control, which may be explained by the fact that a single approach may not be sufficient for such effective control.

Response: We corrected the sentence, as suggested.  

5) “That class” why it was italic.

Response: We corrected the unnecessary italic term.

6) Reference: line 50 check the journal format

Response: We verified the reference according to Pubmed citation and it is correct.  

Lindström P. The physiology of obese-hyperglycemic mice [ob/ob mice].ScientificWorldJournal. 2007 May 29;7:666-85.

7) Reference 68 Not important to cite this paper as this was not a journal. Better to add another reference for this

Response: As suggested, we removed the reference.

Sincerely,

Érika B Rangel, MD, PhD

Reviewer 3 Report

It is a well-design study adding new information to the literature. According to my knowledge, it is a novel paper in its field opening new horizons for further evidence. Authors, succeed to present their findings in a clear way. In addition, the object as well as the results are appropriately discussed in the context of previous literature explaining the importance of the manuscript in its field. Authors succeed to present their data in a clear way adding information to the existing literature.

Therefore, I have no corrections or further work to propose for the improvement of the manuscript and therefore it can be published unaltered.

Author Response

Thank you for supporting our study.